# Production of Biosurfactant Produced from Used Cooking Oil by *Bacillus* sp. HIP3 for Heavy Metals Removal

**DOI:** 10.3390/molecules24142617

**Published:** 2019-07-18

**Authors:** Nurul Hanisah Md Badrul Hisham, Mohamad Faizal Ibrahim, Norhayati Ramli, Suraini Abd-Aziz

**Affiliations:** Department of Bioprocess Technology, Faculty of Biotechnology and Biomolecular Sciences, Universiti Putra Malaysia, 43400 UPM Serdang, Malaysia

**Keywords:** used cooking oil, biosurfactants, *Bacillus* sp. HIP3, oil-degrading bacteria, surfactants

## Abstract

Heavy metals from industrial effluents and sewage contribute to serious water pollution in most developing countries. The constant penetration and contamination of heavy metals into natural water sources may substantially raise the chances of human exposure to these metals through ingestion, inhalation, or skin contact, which could lead to liver damage, cancer, and other severe conditions in the long term. Biosurfactant as an efficient biological surface-active agent may provide an alternative solution for the removal of heavy metals from industrial wastes. Biosurfactants exhibit the properties of reducing surface and interfacial tension, stabilizing emulsions, promoting foaming, high selectivity, and specific activity at extreme temperatures, pH, and salinity, and the ability to be synthesized from renewable resources. This study aimed to produce biosurfactant from renewable feedstock, which is used cooking oil (UCO), by a local isolate, namely *Bacillus* sp. HIP3 for heavy metals removal. *Bacillus* sp. HIP3 is a Gram-positive isolate that gave the highest oil displacement area with the lowest surface tension, of 38 mN/m, after 7 days of culturing in mineral salt medium and 2% (*v*/*v*) UCO at a temperature of 30 °C and under agitation at 200 rpm. An extraction method, using chloroform:methanol (2:1) as the solvents, gave the highest biosurfactant yield, which was 9.5 g/L. High performance liquid chromatography (HPLC) analysis confirmed that the biosurfactant produced by *Bacillus* sp. HIP3 consists of a lipopeptide similar to standard surfactin. The biosurfactant was capable of removing 13.57%, 12.71%, 2.91%, 1.68%, and 0.7% of copper, lead, zinc, chromium, and cadmium, respectively, from artificially contaminated water, highlighting its potential for bioremediation.

## 1. Introduction

Vegetable oils are used globally, especially for food preparation. The most common cooking oil in Malaysia is made from oil palm because of its availability and low price relative to other sources, such as olive, corn, or coconut plants [1]. These vegetable oils are used in cooking due to their contribution to good taste, their attractive colours, and better presentation. As this trend becomes increasingly popular, accumulation of waste generated from cooking oil also increases. There is a growing concern regarding the environmental impact of the rise in production of used cooking oil in homes and restaurants. Kheang et al. (2006) [2] conveyed that around 50,000 tons of used cooking oil, generated from vegetable oils and/or animal fats, is annually disposed of to the environment without proper treatment in Malaysia alone. In the long run, this act contributes to water and soil contamination, causes aquatic life distraction, causes sewer system blockages and overflow, increases water treatment and waste management costs, and consequently generates undesirable impacts to the entire environmental system [3].

The interest in microbial surfactants or biosurfactants has increased steadily in recent years, mainly due to the possibility of production from wastes and their environmentally friendly nature [4]. In addition, the production of biosurfactants from renewable substrates could reduce one of the limiting factors, the high production costs, which are related to incompetent methods for product recovery and purification [5]. Oily substrates such as used cooking oil, which may cause severe environmental problems, have been proven to be good and cheap renewable carbon sources for the production of biosurfactants [6,7,8]. The used cooking oil from domestic waste also contains an appropriate balance of nutrients to support optimum bacterial growth and the synthesis of biosurfactants [7,9].

Biosurfactants have various chemical compositions, generally consisting of fatty acids, glycolipids, lipopeptides, lipopolysaccharides, and lipoproteins, depending on the producing microorganism, the raw materials, and the process conditions. It is an amphiphilic molecule with both hydrophilic and hydrophobic (generally hydrocarbon) moieties that partition preferentially at the interface between fluid phases with different degrees of polarity and hydrogen bonding, such as oil/water or air/water interfaces. Numerous characteristics of biosurfactants, such as foaming, dispersion, wetting, emulsification/de-emulsification, and coating make them suitable to be applied in the physicochemical and biological remediation technologies of both organic and metal contaminants [10].

Heavy metals, which are generally more persistent than organic pollutants due to their non-biodegradability and high toxicity even at trace concentrations, are becoming one of the most serious environmental problems today. They may lead to bioaccumulation in living organisms, causing health problems in animals, plants, and human beings if left untreated in the environment [11]. The property of a microbial surfactant to chelate toxic heavy metals and form an insoluble precipitate was seen as the perfect solution to the treatment of heavy metals in wastewater due to its greater environmental compatibility, lower toxicity, and higher biodegradability than synthetic surfactants [12].

Contaminant sorption relies on the chemical properties of both the soil and the contaminant; hence, the choice of biosurfactant used for contaminant complexation is crucial. The addition of a biosurfactant could promote desorption of heavy metals in two different approaches. The first approach is through the complexation of the free form of metal ions residing in the solution. This would decrease the solution-phase activity of the metal and, therefore, promote desorption, according to Le Chatelier's principle [13]. The second approach is through the accumulation of biosurfactants at the solid-solution interface under the state of reduced interfacial tension. This would permit the direct contact between the biosurfactant and the sorbed metal [14].

One of the most popular biosurfactants used in bioremediation of sites contaminated with toxic heavy metals is surfactin, since it is said to have a powerful surface activity. Surfactin is the cyclic lipopeptide biosurfactant produced by *Bacillus* sp. and it can reduce the surface tension of water from 72 to 27 mN/m at concentrations as low as 0.005% [15]. The potential advantages of using the lipopeptide surfactin include the presence of two charges, due to glutamic and aspartic amino acids, as part of its peptide structure, and thus the binding of heavy metals would be expected [16].

The present work is an initial attempt to systematically screen, isolate, and characterize biosurfactant-producing bacteria capable of utilizing used cooking oil as the sole carbon source. This study also describes the production, characteristics, and surface properties of the extracted biosurfactants of *Bacillus* sp. HIP3, which contributes to the attempts to reduce the biosurfactants’ production cost by using cheap substrates as well as evaluating the efficiency of biosurfactants from used cooking oil, as a substrate by locally isolated bacteria, for heavy metals removal.

## 2. Results and Discussion

### 2.1. Characterization of Used Cooking Oil

Physio-chemical changes that have occurred in used cooking oil typically include changes in colour, odour, viscosity, and compositions. Most of the free fatty acid percentages of the used cooking oil (UCO) used in this study were high when compared with a study conducted by [17] for unused palm oil, except for the palmitic acid. The percentage of palmitic acid in the UCO decreased to 34.5% from 45.6% before it was used. The most highly concentrated fatty acid in the UCO was oleic acid (44.11%), and additionally 5.61% of the oleic acid was introduced into the palm oil after the cooking process. This was followed by the main components in UCO, which are linoleic (12.98%), stearic (4.47%), myristic (1.11%), lauric (0.64%), and linolenic (0.38%). There is no direct relationship between the quality of a used cooking oil and its acid value and it is common to have different fatty acid methyl esters (FAME) percentages for each UCO, as the percentage of FAME increases with the lifetime of the frying oil [18].

### 2.2. Isolation and Screening of Biosurfactant-Producing Bacteria

A total of 14 microorganisms were isolated by a plate and dilution technique from three different sources, which were UCO, cooking oil-contaminated soil, and palm oil mill effluent (POME) sludge. These three sources were selected for microbial isolation due to their richness and diverse microbial community in the natural degradation of oil [19]. The isolated microorganisms comprised six bacterial cultures, three fungi cultures, and five yeast cultures. The isolated microorganisms were screened for the oil-degrader using Bushnell and Haas agar [20] by adding the UCO as the only source of carbon and energy for bacterial growth. Ten out of 14 isolated strains could grow and survive on the agar (Table 1).

From the ten isolated oil-degrading microorganisms, bacterial cultures were selected for further study due to the rapid and abundant growth of the bacteria on the BH agar, as compared to yeast and fungus. Therefore, the microbes were further screened using an oil spreading assay [21]. The results from the oil spreading assay were in complement with the Bushnell and Haas agar screening results, where the positive bacterial strains tested with the agar were also found positive with the oil spreading assay. Morikawa et al. (2000) [21] stated that the oil displacement area in the oil spreading assay is directly proportional to the biosurfactant concentration in the culture broth. Oil displacement occurred at the interface between the two immiscible fluids (oil and water) due to an accumulation of biosurfactants [22]. The accumulation caused the reduction of surface (liquid-air) and interfacial (liquid-liquid) tension. Ultimately, the repulsive forces between the two dissimilar phases reduced, allowing the two phases to mix and interact more easily. Thus, the oil was displaced, forming a clear zone on the water surface. As presented in Table 2, strain HIP3 showed largest oil displacement area of 92.12 cm^2^, hence indicating the highest production of biosurfactant. This is also supported by the lowest surface tension activity of isolate HIP3 after 7 days of cultivation (Table 2).

### 2.3. Identification and Characterization of Biosurfactant-Producing Bacteria

The four biosurfactant producers were subjected to Gram staining in order to characterize the bacterial isolates based on their cell wall constituents between two large groups of bacteria, Gram positive or Gram negative [23]. All the bacterial isolates were Gram positive and rod-shaped and they appeared as purple in colour under the microscope (Table 3). The results obtained were similar to the morphological features of the strain RT10 (*Bacillus siamensis*), which was isolated by Varadavenkatesan & Murty (2013) [24] and wherein it was revealed as rod-shaped and Gram positive bacteria. While the marine isolate, *Bacillus licheniformis* MTCC 5514, was proven as rod-shaped, Gram positive bacteria, which can solubilize or remove crude-oil contamination from different soils in an aqueous phase [25]. This is similar to Zhou et al. (2015) [26], who found out that *Bacillus* sp. ZG0427 was a Gram-positive and spore-forming bacterium whose morphological and biochemical properties were closely related to species of genus *Bacillus*. Analysis of the 16S rRNA gene sequencing showed that all the bacterial strains isolated from UCO and POME sludge belong to the same phylum of Firmicutes with the class of Bacilli. However, they came from a different family and genus (Table 2). According to Tarntip & Sirichom (2011) [27], *Bacillus* and *Lysinibacillus* are ubiquitous and varied either in soil or the environment. Wu et al. (2009) [28] stated that these genera have the ability to catabolize many natural and xenobiotic compounds, such as polynuclear aromatic hydrocarbons.

The phylogenetic trees for all four bacterial isolates are illustrated in Figure 1. The phylogenetic trees were constructed, in which the length of branch shown represents the amount of genetic change of 0.01 and 0.02, respectively. Therefore, the longer the branch in the horizontal dimension, the bigger the amount of change. The unit of branch length used in these phylogenetic trees is percentage (%), which represents the number of changes per 100 nucleotides. In order to find the root for the phylogenetic tree, a sequence that is acknowledged to lie outside of the range of the sequences of interest is included in the data set [29]. Usually, this sequence is referred to as the outgroup. The outgroup used was different for every bacterial isolate; for HIP1, the outgroup used was *Staphylococcus aureus*; for HIP2, the outgroup was *Nocardia sienata*; for HIP3, the outgroup was *Desulfibacterium dehalogenans*; and for HIO1, the outgroup was *Bavariicoccus seileri*.

Research done by Li et al. (2014) [30] shows the strain GSS03T was placed in a cluster within the genus *Bacillus* based on the 16S rRNA gene sequences (1460 bp). These data, which were obtained using the same universal bacterial primer set, 27F and 1492R as in this study, also suggest that strain GSS03T represents a novel species of the genus *Bacillus*. In addition, the 1500 bp PCR fragment of each strain, HIP1, HIP2, HIP3, and HIO1 in this study, display comparable result with *Bacillus* sp. NB22, which had a 1463 bp PCR fragment when 27F and 1492R primers were used [31]. The complete sequence (1433 bases) of the 16S rRNA gene of strain SC-N012T allowed it to be described as a member of a novel species of the genus *Bacillus*, for which the name *Bacillus rhizosphaerae* sp. nov. is proposed [32]. The 16S rRNA gene of strain SC-N012T was amplified from the DNA extract using universal primers 27F and 1492R.

A research done by Peng et al. (2009) [33] stated that *Lysinibacillus* sp. can survive under extremely harsh conditions, making them one of the perfect candidates for the bioremediation of a contaminated environment. As studied by Al-Bahry et al. (2013) [34], members from the genus *Bacillus* are considered as a group suitable for the industrial production of biosurfactants because the species within this taxon, such as *Bacillus licheniformis* and *Bacillus subtilis*, are well known as the producers of surface active metabolites. Furthermore, Neves et al. (2007) [35] stated that the *Bacillaceae* family does not only synthesize good biosurfactants, but also has the ability to grow under anaerobic or facultative environments.

### 2.4. Production of Biosurfactant from Used Cooking Oil

*Bacillus* sp. HIP3 was further selected for biosurfactant production based on its highest oil displacement area and the lowest surface tension measurement. The cell-free culture broth, from the fermentation of the *Bacillus* sp. HIP3, and UCO, as a substrate, was acidified to pH 2 at 4 °C prior to the extraction process, to allow the formation of precipitates (crude biosurfactant). The maximum amount of biosurfactant was achieved using chloroform:methanol (2:1) (5.35 g/L) as the extraction solvent, followed by ethyl acetate (1.86 g/L) and methanol (1.50 g/L), respectively. These results indicated that chloroform:methanol is the suitable solvent to purify the biosurfactants.

The growth phase and surface activity of *Bacillus* sp. HIP3 in MSM with 2% (*v*/*v*) UCO as the sole carbon substrate, at 30 °C and 200 rpm for 7 days, is presented in Figure 2. The highest cell biomass production, 0.95 g/L, was observed at the cultivation time of 156 h. The decrease in biomass production after 6.5 days of fermentation is probably due to growth cessation, which might be caused by the nutritional stress at the end of exponential phase [36]. Earlier, Chandankere et al. (2013) [36] reported nearly the same growth pattern of biosurfactant-producer *Bacillus methylotrophicus* USTBa, isolated from a petroleum reservoir. Their results showed that the biomass production of isolate USTBa had decreased abruptly after 8 days of cultivation using old sunflower frying oil as a substrate. These outcomes revealed that the biosurfactant biosynthesis from UCO occurred predominantly throughout the exponential growth phase, suggesting that the biosurfactant is a primary metabolite and produced accompanying cellular biomass formation due to the production of growth-associated kinetics [36,37].

This study used the acid precipitation and solvent extraction methods to obtain the partially purified biosurfactant as these methods are easy, inexpensive, and readily available to recover crude biosurfactants with low molecular weights, such as lipopeptides and glycolipids from *Bacillus* sp. and *Pseudomonas* sp., respectively [38]. The purpose of all extraction techniques is to separate cellular or fluid lipids from the other constituents, proteins, polysaccharides, and small molecules, hence the process requires the use of very pure solvents and clean glassware. The most widely used solvents are mixtures of chloroform and methanol in a number of ratios, which facilitates adjustment of the polarity of the extraction agent to the target extractable material [39,40]. Chloroform is a popular solvent, particularly for lipids of intermediate polarity, and when combined with methanol it becomes a universal extraction solvent. Furthermore, Shaligram and Singhal (2010) [41] stated that surfactin, a lipopeptide biosurfactant produced by *Bacillus* sp., is acceptable to be extracted with organic solvents as the structure was not destroyed after the extraction process. 

Biosurfactants have a tendency to accumulate at the interface of the medium with air once they are produced in the fermentation broth [42]. Thus, many fractionation processes use some form of separation, in which solutes with high surface activity (biosurfactant) are preferentially adsorbed at the interface between a gas phase and bulk liquid phase and are then removed, for example by foaming [36,43,44]. However, solutes with lower surface activity tend to remain in the bulk aqueous phase and can be extracted by the use of organic solvents [42,45].

An efficient biosurfactant-producing and crude-oil emulsifying bacterium, *B. methylotrophicus* USTBa, was reported to reach the maximum biosurfactant yield of 1.8 g/L using chloroform and methanol (2:1) as extraction solvents [36]. Swapna et al. (2016) [46] obtained 10 g/L of crude biosurfactant by *B. subtilis* SHB 13 from a production medium containing 2% (*w*/*v*) sucrose as a carbon source after extraction using chloroform and methanol as the solvents. The same extraction method was used by Thavasi et al. (2008) [47] to extract maximum glycolipid (7.8 g/L) from *B. megaterium*, using peanut oil cake as substrate. Fermentation studies by Pereira et al. (2013) [48] using LB medium supplemented with 10% (*v*/*v*) sucrose as a carbon source resulted in a maximum biosurfactant yield of 2.16 g/L from *B. subtilis*, isolated from Brazilian crude oils.

### 2.5. Analysis of Biosurfactant

The chromatograms of the biosurfactant from *Bacillus* sp. HIP3, as shown in Figure 3a, revealed the same peaks/fractions with the standard surfactin (Sigma-Aldrich, St. Louis, MI, USA), with the only exception of one peak at a retention time of 65.13 min (Figure 3b). *Bacillus* species produce different types of surface active peptides [49], which are purified from a cell-free supernatant. Other than iturin and fengycin, surfactin is one of the low molecular weight lipopeptides which is synthesized by the *Bacillus* species. Lipopeptides by the *Bacillus* species have a good heterogeneity in accordance with the type and sequence of amino acid moiety and the nature, length, and branching of fatty acid chains and their moiety [50]. Surfactin produced by the *Bacillus* species is the most effective biosurfactant, reducing water surface tension from 72 to 27 mN/m [38,41,49].

The several peaks in Figure 3a show that the biosurfactant that was produced demonstrated the main characteristic groups of a surfactin molecule, indicating the presence of aliphatic hydrocarbon, as well as a peptide fraction [51]. Anburajan et al. (2014) [52] also reported the presence of an additional peak in the chromatogram of the biosurfactant sample from *B. subtilis* 20B, which was not observed in chromatogram of standard surfactin. The retention time of the authentic surfactin standard correlated well with the additional peak obtained from the test samples. HPLC analysis conducted by Sousa et al. (2014) also showed similar retention peaks of biosurfactants produced by *B. subtilis* LAMI005 compared to the commercial surfactin (Sigma-Aldrich, USA). The surfactin groups of compounds are shown to be cyclic lipoheptapeptides, which contain a,β-hydroxy fatty acid in its side chain [41]. Natural surfactins are a mixture of isoforms A, B, C, and D and are classified in accordance with the difference in their fatty acid sequence. They are cyclic lipopeptides and are heptapeptides interlinked with a,β-hydroxy fatty acid. Hence, it was suggested that the additional fraction/peak shows no significant difference between the chromatogram of the biosurfactant sample and the commercial surfactin, as it is not part of surfactin’s isoforms [51].

### 2.6. Heavy Metals Removal

Biosurfactant mediated metal chelation was observed by treating 10 mg/mL of surfactin with a lead, copper, zinc, chromium, and cadmium solution over a period of 24 h at room temperature [46]. Based on Figure 4, the biosurfactant was able to remove 13.57%, 12.71%, 2.91%, 1.68%, and 0.7% of 100 ppm of copper, lead, zinc, chromium, and cadmium, respectively. The precipitation was caused due to charge neutralization upon either the addition of the metal cations into the anionic biosurfactant solution or vice versa [13,53,54].

Singh and Cameotra (2013) [55] claimed to remove 26.2% copper and 32.1% zinc by using 1.5 mg/mL of a mixture of biosurfactants containing surfactin and fengycin. On the other hand, this study demonstrates the removal of 13.6% copper and 2.9% zinc using only surfactin as the biosurfactant produced by *Bacillus* sp. HIP3. During the treatment, the positively charged metals bind to the outer hydrophilic surface of the biosurfactants, which have anionic peptide head groups to remove the contaminant [56].

This study also shows the potential of removing 12.7% lead and 0.7% cadmium, which is lower than the studies reported by Das et al. (2009) [56] and Singh and Cameotra (2013) [55], which can remove 100% lead and 97.66% cadmium and 40.3% lead and 44.2% cadmium, respectively. However, these researchers used lipopeptide biosurfactants comprising more than one type of biosurfactant, which explains the higher percentage removal of the metals. This is also supported by Lima et al. (2011) [57], as they successfully removed 99.26% cadmium by using 0.3 mg/mL of lipopeptide biosurfactants, namely surfactin, iturin, and fengycin, produced by *B. subtilis* LBBMA 111A.

According to Maikudi et al. (2016) [13], the selectivity of biosurfactants for metals, both in solution and in soil systems, must be examined. Since this result was demonstrated based on the preliminary study on the capability of surfactin produced by *Bacillus* sp. HIP3 for heavy metals removal, a further study will be required to find the information on the biosurfactant structure and size, or the efficacy of biosurfactant-metal interactions on these structures. In addition, the biosurfactant structure charge and size can influence the movement of biosurfactant-metal complexes through the solution and predict the fate of the heavy metals in the environment [14].

## 3. Materials and Methods

### 3.1. Characterization and Preparation of Raw Material/Substrate

Used cooking oil is the renewable raw material used in this study. The analysis of fatty acid compositions of the UCO was done using gas chromatography–mass spectrometry (GC–MS). The UCO must be prepared in the form of methyl esters by sodium methoxide method prior to analysis [58]. A volume of 5 mL of hexane was added into a vial containing 100 mg of UCO and it was vortexed briefly to dissolve the oil. A volume of 250 µL of sodium methoxide reagent was added into the vial and the vial was closed tightly prior to a vortex for 1 min (with a pause every 10 s). Next a volume of 5 mL of saturated sodium chloride was added, the vial was closed tightly, and shaken vigorously for 15 s. It was left for 10 min. The hexane layer was removed and transferred to the vial containing a small amount sodium sulphate (any aqueous phase interfacial precipitate was not transferred). The hexane phase containing the methyl esters was allowed to contact with sodium sulphate for at least 15 min prior to analysis. Used cooking oil as the substrate for biosurfactant production was prepared by filtering, using a sieve to remove large particles and suspended materials. The UCO was then sterilized using a 0.22 µm sterile filter and a 50 mL sterile syringe. It was stored at room temperature prior to fermentation.

### 3.2. Isolation and Screening of Biosurfactant-Producing Bacteria

Three types of sources were used for the isolation of biosurfactant-producing bacteria, including used cooking oil (UCO), oil contaminated soil, and palm oil mill effluent (POME) sludge. The direct isolation of the biosurfactant producer was performed by diluting and plating the sample on nutrient agar plates [59]. After 48 h of incubation at 37 °C in an inverted condition, a loopful of colonies with the same morphological characteristics was streaked onto new nutrient agar plates. This sub-culturing step was repeated until pure single isolates were obtained. 

The pure cultures obtained were then screened using Bushnell and Haas (BH) agar, in order to find the oil-degrading microorganisms, by adding UCO as the sole carbon source onto the agar [20]. The composition of Bushnell and Haas agar (Atlas, 2010) in grams per liter is as follows: KH_2_PO_4_ (1.0), K_2_HPO_4_ (1.0), NH_4_NO_3_ (1.0), MgSO_4_.7H_2_O (0.2), FeCl_3_ (0.05), CaCl_2_.2H_2_O (0.02), and agar (15.0). After the inoculated agar was prepared, 100 µL of sterile UCO was smeared on top of the agar as the sole carbon source for the microorganisms and was incubated at 37 °C for 48 h. The oil-degrading microorganisms were identified based on their ability to deteriorate and grow on the BH agar. The selected oil-degrading microorganisms were sub-cultured onto nutrient agar and incubated for future use. 

In the oil spreading assay, a volume of 50 mL of distilled water was placed into a large petri dish (25 cm diameter) followed by the addition of 20 µL of crude oil to the center of the plate. Then, 10 µL of culture broth were added on top of the oil surface on separate plates [21]. The culture was prepared in mineral salt medium (MSM), from the adjusted methods of Oliveira and Garcia-Cruz (2013) [7], and maintained at 30 °C with an agitation of 200 rpm in a rotary shaker for 7 days. As a method of measurement for the oil spreading assay, the diameter of the clear zone on the surface of the oil was measured. From the average diameter obtained, the oil displacement area was determined using the formula for the area of a circle which is πr^2^.

### 3.3. Preparation of Culture Medium

The mineral salts medium (MSM) used in this study was modified from Oliveira and Garcia-Cruz (2013) [7]. Conical flasks, measuring 250 mL, were filled with 50 mL of MSM with the following composition (g/L): NaNO_3_ (7.0), KH_2_PO_4_ (0.5), K_2_HPO_4_ (1.0), KCl (0.1), MgSO_4_·7H_2_O (0.5), CaCl_2_·2H_2_O (0.01), FeSO_4_·7H_2_O (0.01), and yeast extract (0.1). The medium was adjusted to pH 7 with 1 N NaOH and sterilized at 121 °C for 15 min prior to the addition of 2% (*v*/*v*) sterile used cooking oil (carbon source) and 2% (*v*/*v*) inoculum. The preparation of the inoculum was done by adding a single colony of a bacterial strain in a nutrient agar plate into 100 mL of sterile nutrient broth. The broth was then incubated at 37 °C and agitated at 200 rpm until the OD_600nm_ reached 0.60 to 1.00.

### 3.4. Characterization and Identification of Biosurfactant-Producing Bacteria

Out of four oil-degrading bacteria, the bacterial strain whose culture broth was determined to have the highest oil displacement area and the lowest surface tension value was selected as the biosurfactant producer. To characterize the biosurfactant-producing bacteria, the Gram of the isolate as well as its morphologies, such as shape and colour, were determined [60]. The growth profile of the strain was also determined by transferring a loopful of a colony into 100 mL of nutrient broth, which was subsequently incubated at 37 °C in a rotary shaker and agitated at 200 rpm for 48 h.

The genomic DNA of the selected biosurfactant producer was used to determine the taxonomic characterization of the strain using 16S rRNA gene sequencing analysis. Total DNA was extracted from 2 mL of pure culture grown overnight in a nutrient broth at 37 °C and agitated at 200 rpm using an UltraClean® Microbial DNA Isolation Kit (MO BIO Laboratories Inc., Calsbad, CA, USA). The bacterial 16S rRNA loci were amplified using universal forward primer JCM27F (5′-AGAGTTTGATCCTGGCTCAG-3′) and universal reverse primer JCM1492R (5′-GGTTACCTTGTTACGACTT-3′). The 20 µL reaction mixture contained 0.5 µL of DNA template, 2.0 µL of 10 × PCR buffer, 1.2 µL of 1.5 mM MgCl_2_, 0.1 µL of 5 U/µL *Taq* polymerase, 0.5 µL of 10 mM dNTP, 0.5 µL of each forward and reverse primer, and 14.7 µL of distilled water. The temperature profile for PCR was 94 °C for 5 min (1 cycle), 94 °C for 1 min, 55 °C for 45 s, and 72 °C for 45 s (30 cycles); and 72 °C for 1 min after the final cycle [61]. It was carried out on the TaKaRa PCR Thermal Cycler Dice (model TP600, Takara Bio Inc., Otsu, Shiga, Japan) with a running process of 2.5 h. The sequencing of purified products was performed by First Base Laboratories Sdn. Bhd. (Selangor, Malaysia). The nucleotide sequence of the 16S rRNA gene of the isolate was compared with all deposited nucleotide sequences in GenBank database using the BLAST program on the National Center for Biotechnology Information (NCBI) website (http://blast.ncbi.nlm.nih.gov). The alignments were analyzed to construct a phylogenetic tree and to compare similarities among the sequences by the neighbor-joining method using MEGA software version 7.0 (Pennsylvania State University, State College, PA, USA).

### 3.5. Production of Biosurfactant from Used Cooking Oil as Carbon Source

The locally isolated bacterial strains, which are designated as HIP1, HIP2, HIP3, and HIO1, were transferred to 100 mL nutrient broth (Merck, Darmstadt, Germany) and incubated at 30 °C and agitated at 150 rpm in an incubator shaker (Labwit, model ZHWY-1102C, Victoria, Australia) until reaching the exponential phase, based on optical density at 600 nm, as a seed culture. Each of the bacterial suspensions (2%, *v*/*v*) were inoculated into a 250 mL Erlenmeyer flask containing 50 mL of mineral salt medium (MSM) [7] and 2% (*v*/*v*) of UCO, respectively. The initial pH of the medium was adjusted to 7.0 prior to sterilization by an autoclave (HICLAVE, model HVE-50, Mumbai, India). It was then incubated in a rotary incubator shaker (200 rpm) at 30 °C for 7 days. Samples of culture medium were taken every 24 h and stored at 4 °C until processing for the cell biomass and surface tension estimation.

### 3.6. Analytical Methods

#### 3.6.1. Cell Biomass Determination

A volume of 2 mL of culture broth samples, which were collected at periodic time intervals in a sterile manner, were centrifuged at 10,000 rpm for 20 min (4 °C). The biomass pellet was washed thrice with a 0.9% *w*/*v* saline solution. The paste was dried by heating in a hot air oven set at 50–70 °C until a constant weight was attained, without allowing the cells to be charred [24].

#### 3.6.2. Surface Activity Measurements

The surface tension of cell-free culture broth (previously centrifuged at 10,000 rpm and 4 °C for 20 min) was measured by the Du Nuoy–Padday method [62] using an AquaPi^+^ model tensiometer (Kibron, Helsinki, Finland). The platinum dyne probe and glassware were cleaned with deionized water and acetone prior to measurement. A volume of 2 mL of the broth was transferred into a new and clean dyne cup and placed onto the tensiometer platform. Then, the platinum wire rod was submerged into the broth and then slowly pulled through the liquid–air interface to measure the surface tension (mN/m). Between each measurement, the platinum wire rod was rinsed with water, flamed, and was allowed to dry. All the measurements were taken in triplicate. The calibration was done using deionized water (ST = 71.5 ± 0.5 mN/m) before taking samples measurement.

### 3.7. Extraction of Biosurfactant

The crude biosurfactant molecules produced by *Bacillus* sp. HIP3 were obtained primarily by the acid precipitation method. The culture broth was centrifuged at 10,000 rpm and 4 °C for 20 min to obtain a cell-free supernatant. The pH of the cell-free supernatant was adjusted to 2.0 with 3 M HCl. This solution was maintained at rest for 12 h (overnight) at 4 °C to allow the precipitation of the biosurfactant. It was then centrifuged at 10,000 rpm and 4 °C for 15 min. The supernatant was removed and the pellet was extracted using several analytical grade solvents (>99.9% purity), namely methanol [63], ethyl acetate [61], and 2:1 chloroform-methanol [24], respectively, by continually stirring for 2 h in order to find the best solvent for the biosurfactant extraction. It was then evaporated to dryness using a rotary evaporator (Buchi, model Rotavapor R-210, Flawil, Switzerland), leaving behind the relatively pure biosurfactant as a viscous light brown matter. It was weighed to record the amount and stored at 4 °C for further analysis [63].

### 3.8. Analysis of Biosurfactant

The basic steps in high-performance liquid chromatography (HPLC) are as follows: A semi-preparative HPLC column, which was a Phenomenex Luna C18 (250 mm × 10 mm id), was connected to the HPLC. The sample was prepared by dissolving in water/TFA (99.95:0.05, *v*/*v*). It was centrifuged at 13,000 rpm for 5 min to remove particulate matter prior to injection to the HPLC at a wavelength of 210 nm. Mobile phase A (water/TFA (99.95:0.05, *v*/*v*)) and B (acetonitrile/water/TFA (80:19.95:0.05, *v*/*v*)) were prepared and a gradient elution was performed starting with 100% A, 0% B and changing to 0% A, 100% B over 80 min [64]. The flow rate and injection volume were set at 1.0 mL/min and 500 µL/min, respectively.

### 3.9. Heavy Metals Remediation Efficiency

The ability of the biosurfactant to remove heavy metal was evaluated using 100 ppm of heavy metals in their salt forms. The medium without biosurfactants and organisms served as the control. The biosurfactant (surfactin) was recovered after 7 days in the cell free culture broth using a solvent extraction method [63] and 10 mg of the dried extracted biosurfactant was dissolved in 10 mM Tris buffer (pH 7.2). It was then added into 100 mL of a 100 ppm copper, lead, zinc, chromium, and cadmium solution (Sigma-Aldrich, USA), respectively, and mixed well prior to incubation at 30 °C in a shaker incubator. After 72 h of incubation, these solutions were drawn and centrifuged to separate the metal biosurfactant complex (precipitate) formed and the supernatant containing unbound metal ions. The samples were then analyzed for the concentration of metals present after treatment according to standard procedure using atomic absorption spectroscopy (Thermo Fisher Scientific, model iCE™ 3300, Waltham, MA, USA) [46]. All experiments were performed in triplicate, and the average of the results is presented. The metal removal efficiency was calculated using the formula in Equation (1) [57], as follows:(1)$$Percent\;removed=\;\frac{Amount\;of\;heavy\;metals\;in\;supernatant}{Total\;amount\;of\;heavy\;metals}\;\times 100\%.$$

## 4. Conclusions

The locally isolated bacteria *Bacillus* sp. HIP3 produced 5.35 g/L of surfactin from used cooking oil as the sole carbon source. The lowest surface tension value of supernatant from *Bacillus* sp. HIP3, when cultivated in mineral salt medium (MSM) with 2% (*v*/*v*) UCO at 30 °C and 200 rpm, was 38.15 mN/m at 156 h of cultivation. The surfactin is capable of removing 13.57%, 12.71%, 2.91%, 1.68%, and 0.7% of 100 ppm of copper, lead, zinc, chromium, and cadmium, respectively from an artificially contaminated solution. The property of this surfactin to remove heavy metals, by forming an insoluble precipitate, may find numerous applications in treatment of wastewater containing heavy metals.

## Figures and Tables

**Figure 1 molecules-24-02617-f001:**
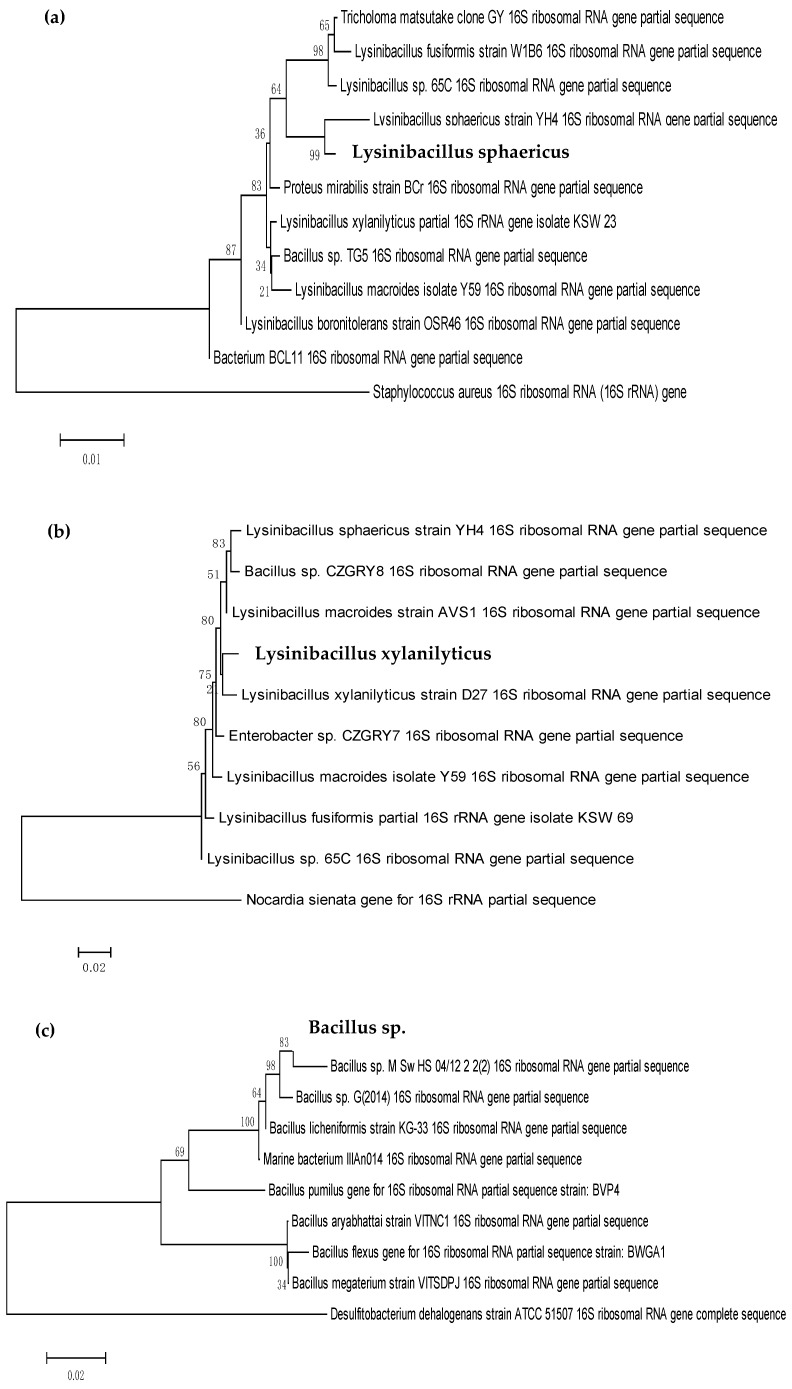
Evolutionary relationships of taxa of isolate (**a**) HIP1, (**b**) HIP2, (**c**) HIP3, and (**d**) HIO1. The evolutionary history was inferred using the neighbor-joining method. The percentage of replicate trees in which the associated taxa clustered together in the bootstrap test (100 replicates) is shown next to the branches. The tree is drawn to scale, with branch lengths in the same units as those of the evolutionary distances used to infer the phylogenetic tree. The evolutionary distances were computed using the Maximum Composite Likelihood method and are in the units of the number of base substitutions per site. The analysis involved 37 nucleotide sequences. All positions containing gaps and missing data were eliminated. Evolutionary analyses were conducted in MEGA5.

**Figure 2 molecules-24-02617-f002:**
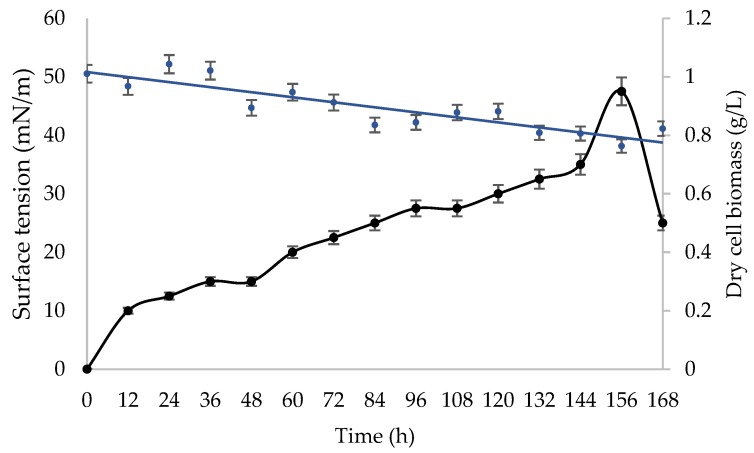
Cell biomass and surface activity profile of *Bacillus* sp. HIP3 when cultivated in MSM with 2% (*v*/*v*) UCO as a carbon source at 30 °C and 200 rpm for biosurfactant production.

**Figure 3 molecules-24-02617-f003:**
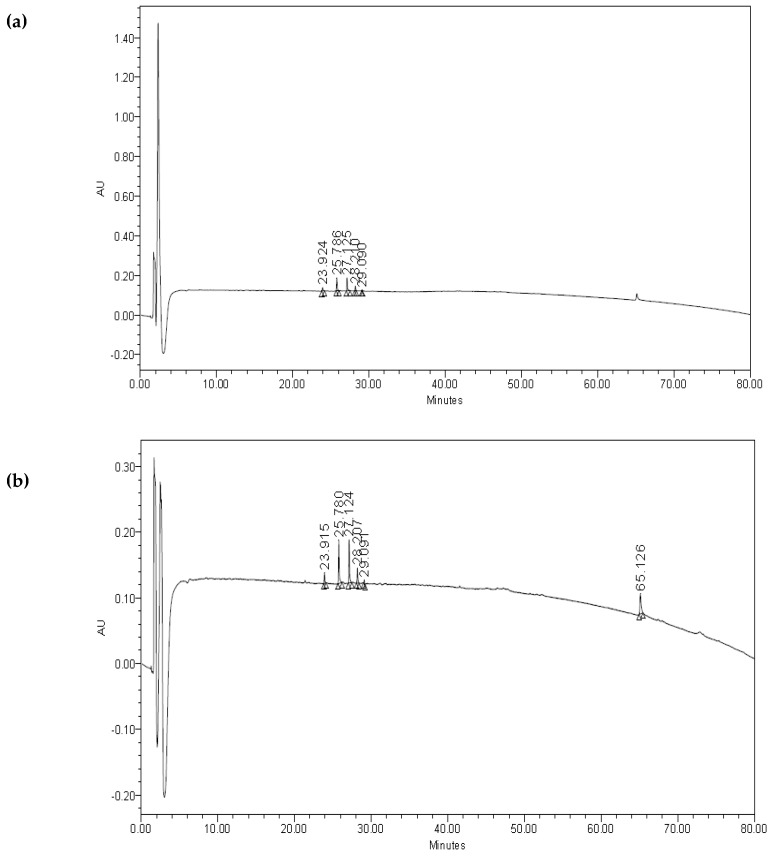
High performance liquid chromatography chromatograms of the biosurfactant produced by (**a**) *Bacillus* sp. HIP3 and (**b**) the standard Surfactin (Sigma-Aldrich, USA).

**Figure 4 molecules-24-02617-f004:**
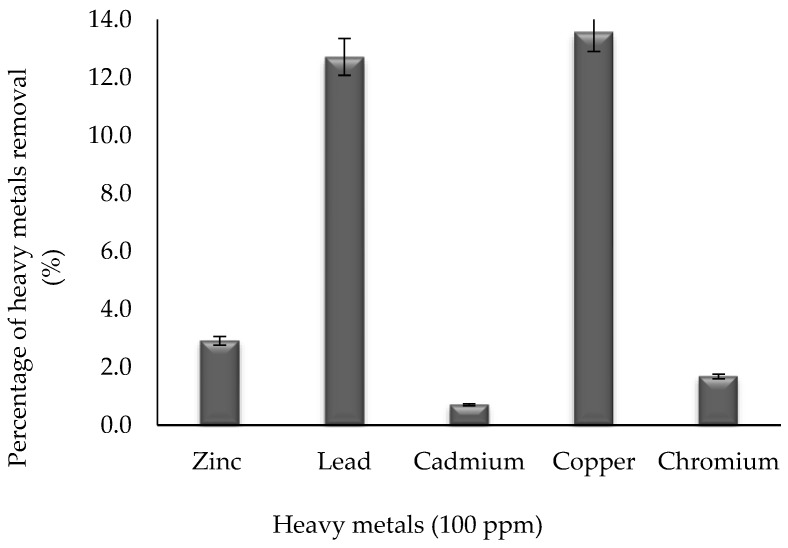
Heavy metals removal using surfactin produced from *Bacillus* sp. HIP3 after treatment for 24 h.

**Table 1 molecules-24-02617-t001:** Microorganisms screened using Bushnell and Haas agar.

Microorganisms	Isolate	Source
Bacteria	HIP1	POME
	HIP2	POME
	HIP3	POME
	HIO1	UCO
Yeast	HIP4	POME
	HIP5	POME
	HIP6	POME
	HIS1	Cooking oil contaminated soil
Fungi	HIS2	Cooking oil contaminated soil
	HIS3	Cooking oil contaminated soil

**Table 2 molecules-24-02617-t002:** Oil spreading assay and surface tension measurements of potentially biosurfactant producing bacteria.

Isolates	Oil Displacement Area (cm^2^)	Surface Tension (mN/m)
HIP1	33.18	43.5
HIP2	29.9	43.33
HIP3	92.12	38.15
HIO1	26.7	50.17

**Table 3 molecules-24-02617-t003:** Gram staining and identification of the bacterial isolates by 16S rRNA gene sequence analysis.

Isolate	Shape	Gram	Genus/Species	Similarity (%)
HIP1	Rod	Positive	*Lysinibacillus sphaericus* (KJ576904.1)	99
HIP2	Rod	Positive	*Lysinibacillus xylanilyticus* (LK391655.1)	75
HIP3	Rod	Positive	*Bacillus* sp. (AY787805.1)	83
HIO1	Rod	Positive	*Bacillus* sp. (KC160846.1)	58

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
