# Peer review of "Production of Biosurfactant Produced from Used Cooking Oil by Bacillus sp. HIP3 for Heavy Metals Removal"

_molecules, 2019, doi:10.3390/molecules24142617_

Round 1

Reviewer 1 Report

The authors have isolated biosurfactant producing microorganisms, they produced biosurfactant, carried out characterization  and   examined its potential application for heavy metal remediation from contaminated water.

Generally the authors have done a work that has novelty but it has many drawbacks, their  English is  poorly written, and some characterizations of the biosurfactant  are lacking . Thus I suggest acceptance of the paper with major revision hoping that they would address my comments seriously

Some of my comments are

Comment 1

Their English  has lots of  typogrammatical  and consistency, punctuation and clarity problems thus,  the whole text  needs to be  revised and corrected by fluent English speakers,

Comment 2

The HPLC  biosurfactant characterization with reference to commercial surfactine is  good but it  is not enough  on its own,  thus they need to add some additional characterizations to confirm that the biosurfactant is surfactine. I suggest that  the physicochemical characterization they did for the cooking oil should be done for the biosurfactant as well,

The authors have isolated biosurfactant producing microorganisms, they produced biosurfactant, carried out characterization  and   examined its potential application for heavy metal remediation from contaminated water.

Generally the authors have done a work that has novelty but it has many drawbacks, their  English is  poorly written, and some characterizations of the biosurfactant  are lacking . Thus I suggest acceptance of the paper with major revision hoping that they would address my comments seriously

Some of my comments are

Comment 1

Their English  has lots of  typogrammatical  and consistency, punctuation and clarity problems thus,  the whole text  needs to be  revised and corrected by fluent English speakers,

Comment 2

The HPLC  biosurfactant characterization with reference to commercial surfactine is  good but it  is not enough  on its own,  thus they need to add some additional characterizations to confirm that the biosurfactant is surfactine. I suggest that  the physicochemical characterization they did for the cooking oil should be done for the biosurfactant as well,

The authors have isolated biosurfactant producing microorganisms, they produced biosurfactant, carried out characterization  and   examined its potential application for heavy metal remediation from contaminated water.

Generally the authors have done a work that has novelty but it has many drawbacks, their  English is  poorly written, and some characterizations of the biosurfactant  are lacking . Thus I suggest acceptance of the paper with major revision hoping that they would address my comments seriously

Some of my comments are

Comment 1

Their English  has lots of  typogrammatical  and consistency, punctuation and clarity problems thus,  the whole text  needs to be  revised and corrected by fluent English speakers,

Comment 2

The HPLC  biosurfactant characterization with reference to commercial surfactine is  good but it  is not enough  on its own,  thus they need to add some additional characterizations to confirm that the biosurfactant is surfactine. I suggest that  the physicochemical characterization they did for the cooking oil should be done for the biosurfactant as well,

Reviewer 2 Report

The manuscript with title "Production of Biosurfactant Produced from Used Cooking Oil by Bacillus sp. HIP3 for Heavy Metals Removal" is aimed at screening microorganisms for production of biosurfactant from cheap carbon source, and at evaluatig the possible application in heavy metals removal. The topic is of great interest, because currently the main problem in BS research is the high cost of production. The use of cheap materials could represent the possibility to overcome the problem of the cost production, on the one hand, and the disposal of certain types of wastes, on the other hand.  Overall, the manuscript is well structured, but can be improved in some aspects. Here are some suggestions:

Introduction

The topic is well described and focused. However, the use of BS for heavy metal removal should be a little be improved. Several works describe the heavy metal removal from solid and liquid matices, with some differences. This aspect should be treated deeply in the introductiond, as well as in the discussion section.

Line 74: Bacillus should be written in italics

Line 81: Here the authors refer to 'biorurfactants', but in reality it has been investigated the effeciency of only Bacillus sp. HIP3 biosurfactant. Please specify better.

Results and Discussion

Lines 90-91. Please adjust the sentence. It is not clear.

Line 94: Please specify 'FAME'

Line 98: Please specify 'POME'

Lines 100-101. I think the English form needs to be enhanced.

Lines 114-115. I think the English form needs to be enhanced.

Line 116. I suggest to insert a table with the results of oil displacemente test from all tested microorganisms, to be able to appreciate the result obtained from the isolate HIP3.

Table 2. I think you should provide the accession number of deposited sequences.

Figure 2. I believe this image is not necessary, and could be eliminated. 

Line 183-185. This point is not clear at all. The surface tension measurements have been performed as selective screening procedure, together with the oil displacement test? If yes, this part should be moved with the results of oil displacement test, and the results of both screening tests should be inserted in a table. 

Lines 186-189. This part should be moved in the material and method section.

Line 192: Please adjust the English form.

Lines 213-2018. I suggest to move this part a little later, to motivate and contextualize the choice of the extracion methods.

Line 232: Replace biosurfactant production with 'biosurfactant yields'

Line 242: 'With the only excepiton of...' instead of 'except'

Line 271: I think a verb is missing.

Lines 279-280: This seems a repetition of results previously described.

Line 283: is it mL/mg?

Line 290: delete the semicolon

Figure 6. Delete 'types of'

Materials and Methods

Lines 309-316. This part should be moved after the description of the screening tests

Line 330: Replace 'its' with 'their'

Line 366: Here the authors assert that all isolates have been cultured to monitor the BS production, but results are showed only for one strain. 

Line 374: delete 'it was' and adjust as follows: 'stored at 4°C until processing with cell biomass and surface tension estimation.

Line 393: Please specify which strain

Lines 413-414: which heavy metal concentrations? Please specify. Results are showed only for 100 ppm. 

The English language should be revised throughout the manuscript, especially with regard to the conjugation of verbs. The conclusions should be deepened and contextualized. In this case the BS-mediated removal of heavy metals is appreciable in a liquid system, while literature is rich of references about removal from solid matrices. I suggest to improve this aspect (some references could be helpful, as such as

Das P, Mukherjee S, Sen R (2009) Biosurfactant of marine origin exhibiting heavy metal remediation properties. Bioresour Technol

100:4887–4890

Rizzo C., Michaud L., Graziano M., De Domenico E., Syldatk C., Hausmann R., Lo Giudice A. (2015). Biosurfactant activity, heavy metal tolerance and characterization of Joostella strain A8  from the Mediterranean polychaete Megalomma claparedei (Gravier, 1906). Ecotoxicology, 24:1294–1304) and to highlight better the use of cheap raw materials.  

Round 2

Reviewer 1 Report

Reviewer’s Comments

The authors have done considerable revision, thus I suggest minor revision for the final acceptance of the paper. I have tried to mention some of my comments that I suggest should be addressed as some errors are still there. I have given some of my comments as follows

Page 3 line 126-127

Based on Table 2, strain HIP3 was shown largest oil displacement area of 92.12 cm2  , hence indicates highest production of biosurfactant

I suggest rewriting it  like   ”  as presented in table 2 strain HIP3  showed largest oil displacement area…………..”

PAGE 13 lines 406-408,, what do you mean by “ ….. removed and the pellet was extracted with several analytical grade solvents (>99.9% purity) namely  methanol [63], ethyl acetate [61], and 2:1 chloroform-methanol [24], respectively for 2 h while stirring  continuously in order to employ the best solvent for biosurfactant extraction????,please rewrite it clearly and succinctly what do you mean by while stirring continuously to employ the best solvent??? It is unnecessary and wrong expression

Page 13 lines  422-423

The ability of the biosurfactant to remove heavy metal was evaluated using 100 ppm of heavy  metal in the form of  salts ??, I suggest  rewriting it   rather as ……in their salt forms..
